# Salivary Metabolic Profile of Patients with Lung Cancer, Chronic Obstructive Pulmonary Disease of Varying Severity and Their Comorbidity: A Preliminary Study

**DOI:** 10.3390/diagnostics10121095

**Published:** 2020-12-15

**Authors:** Lyudmila V. Bel’skaya, Elena A. Sarf, Denis V. Solomatin, Victor K. Kosenok

**Affiliations:** 1Biochemistry Research Laboratory, Omsk State Pedagogical University, 14, Tukhachevsky str, 644043 Omsk, Russia; nemcha@mail.ru; 2Department of Mathematics and Mathematics Teaching Methods, Omsk State Pedagogical University, 14, Tukhachevsky str, 644043 Omsk, Russia; denis_2001j@bk.ru; 3Department of Oncology, Omsk State Medical University, 12, Lenina str, 644099 Omsk, Russia; victorkosenok@gmail.com

**Keywords:** lung cancer, chronic obstructive pulmonary disease, saliva, biochemistry, diagnostics, prognosis

## Abstract

The aim of the work was to study the features of the salivary biochemical composition in the combined pathology of lung cancer and chronic obstructive pulmonary disease (COPD) of varying severity (COPD I, COPD II). The study group included patients with lung cancer (*n* = 392), non-malignant lung pathologies (*n* = 168) and healthy volunteers (*n* = 500). Before treatment, the salivary biochemical composition was determined according to 34 indicators. Survival analysis performed by the Kaplan-Meier method. Biochemical parameters (catalase, imidazole compounds ICs, sialic acids, lactate dehydrogenase (LDH)) that can be used to monitor patients at risk (COPD I) for timely diagnosis of lung cancer are determined. A complex of salivary biochemical indicators with prognostic value in lung cancer was revealed. For patients with lung cancer without COPD, a group of patients with a favorable prognosis can be distinguished with a combination of ICs < 0.478 mmol/L and LDH >1248 U/L (HR = 1.56, 95% CI 0.40–6.07, *p* = 0.03891). For COPD I, a level of ICs < 0.182 mmol/L are prognostically favorable (HR = 1.74, 95% CI 0.71–4.21, *p* = 0.07270). For COPD II, combinations of pH < 6.74 and LDH >1006 U/L are prognostically favorable. In general, for patients with lung cancer in combination with COPD I, the prognosis is more favorable than without COPD.

## 1. Introduction

Lung cancer is one of the predominantly diagnosed cancers (11.6% of the total number of cases); it was the leading cause of cancer death in 2018 (18.4% of the total number of cancer deaths) [1,2]. Another common lung disease is a chronic obstructive pulmonary disease (COPD) [3]. Recently, COPD has become the focus of attention due to increased morbidity, mortality and increased risk of lung cancer [4]. COPD and lung cancer have common features: high mortality and risk factors such as smoking, some genetic background, environmental exposure and major common inflammatory processes [5]. Several studies have shown that COPD is a risk factor for lung cancer, regardless of exposure to smoking, with a four to six times increased risk of lung cancer [6]. This risk increases with a progressive decline in FEV1, regardless of smoking history [7]. In addition, COPD worsens the prognosis of lung cancer due to higher morbidity and mortality [8]. There are numerous studies confirming the role of chronic inflammation in the genesis of cancer in general and lung cancer in particular [9]. It was found that various dysregenerative changes in the pulmonary epithelium against the background of chronic inflammatory lung disease have a malignant potential [10]. This explains the fact of frequent development of lung cancer against the background of COPD, in which dysplastic changes and metaplasia of the bronchial epithelium are clinical manifestations. Until now, no biomarker has been identified that would unequivocally determine the presence of lung cancer and/or COPD using both sputum and exhaled air and blood serum [11,12,13]. In previous studies, including ours, we have shown that the composition of saliva reflects metabolic changes occurring against the background of lung cancer [14,15,16]. However, a study of the effects of the combined pathology of lung cancer and COPD on the metabolic profile of saliva has not yet been conducted. The aim of the work was to study the characteristics of the biochemical composition of saliva in combined pathology, lung cancer and COPD of varying severity.

## 2. Materials and Methods

### 2.1. Study Design and Group Description

The study included 593 patients hospitalized in the thoracic department of the Clinical Oncological Dispensary in Omsk in the period 2014–2017. The inclusion criteria were: the age of the patients 30–75 years, the absence of any treatment, including surgery, chemotherapy or radiation. The collection of saliva samples was carried out strictly before the start of treatment.

A detailed description of the study group is given in Table 1. After histological verification, 168 people (28.3%) were diagnosed with non-malignant lung pathologies, including 51—hamartoma, 30—sarcoidosis, 28—tuberculoma, 39—fibrosis/pneumosclerosis, 13—inflammatory tumor, 4—pneumonia, 2—papilloma, 1—lipoma. These patients made up the comparison group. In 425 patients (71.7%), lung cancer of various histological types was confirmed, including 189—adenocarcinoma (ADC), 135—squamous cell carcinoma (SCC), 8—mixed (ADC + SCC), 68—neuroendocrine cancer (NEC) and 25—undifferentiated lung cancer. The NEC group included 16 patients with a diagnosis of typical and atypical carcinoid (low-grade G1 + G2) and 45 patients with small cell lung cancer, and 7 patients with large cell lung cancer (high-grade G3). Patient groups with mixed and undifferentiated cancers were subsequently excluded from the study. The control group consisted of 500 healthy patients, in whom no lung pathology was detected during routine clinical examination.

To describe the severity of COPD, a classification based on FEV1 as a percentage of predicted was used [17]. GOLD criteria were used to classify the severity of COPD (GOLD I; FEV1 > 80% of predicted, GOLD II; FEV1 = 50–79% of predicted, GOLD III; FEV1 = 30–49% of predicted, and GOLD IV; FEV1 < 30% of predicted). In this work, we examined groups of patients without COPD, with mild and moderate COPD (GOLD I, GOLD II), designated, respectively NO COPD, COPD I and COPD II.

The study was approved at a meeting of the Ethics Committee of the Omsk Regional Clinical Hospital “Clinical Oncology Center” on 21 July 2016 (Protocol No. 15). All of the volunteers provided written informed consent.

### 2.2. Collection, Processing, Storage and Analysis of Saliva Samples

Saliva (5 mL) was collected from all participants prior to treatment. Collection of saliva samples was carried out on an empty stomach after rinsing the mouth with water in the interval of 8–10 am by spitting into sterile polypropylene tubes; the salivation rate (mL/min) was calculated. Saliva samples were centrifuged (10,000× *g* for 10 min) (CLb-16, Moscow, Russia), after which biochemical analysis was immediately performed without storage and freezing using the StatFax 3300 semi-automatic biochemical analyzer [16].

### 2.3. Statistical Analysis

Statistical analysis was performed using Statistica 13.3 EN software (StatSoft, Tulsa, OK, USA); R version 3.6.3; RStudio Version 1.2.5033; FactoMineR version 2.3. (RStudio, version 3.2.3, Boston, MA, USA) by a nonparametric method using the Mann-Whitney *U*-test and the Kruskal-Wallis *H*-test. At the preliminary stage, the character of distribution and homogeneity of dispersions in groups was checked. According to the Shapiro-Wilk test, the content of all determined parameters does not correspond to the normal distribution (*p* < 0.05). The test for the homogeneity of variances in groups (Bartlett’s test) allowed us to reject the hypothesis that variances are homogeneous across groups (*p* = 0.00017). Therefore, nonparametric statistical methods were used to process the experimental data. The description of the sample was made by calculating the median (Me) and the interquartile range as the 25th and 75th percentiles (LQ; UQ). Differences were considered statistically significant at *p* < 0.05.

The Kruskal-Wallis test is a nonparametric alternative to one-dimensional (intergroup) ANOVA. It is used to compare three or more samples. With a high significance of the Kruskal-Wallis test (*H*), the characteristics of different experimental groups significantly differ from each other (*p* < 0.05). Using the Kruskal-Wallis test, we compared several groups (3 groups in Table 2 and Table 3; 5 groups in Table 4) and selected indicators whose change was significant at *p* < 0.05. These indicators were subsequently used for the principal component analysis (PCA). In addition, we included indicators in the PCA analysis, the values of which differ at the 0.10 significance level. In the case of 0.05 < *p* < 0.10, the limit of significance is slightly exceeded, which means that there is a tendency towards the manifestation of a pattern. If a significant pattern is identified, to identify groups that are significantly different from each other, it is necessary to test all groups in pairs. The Mann-Whitney test was used only for pairwise comparison of the differences between the COPD and NO COPD groups in different histological types of lung cancer; in all other cases, the Kruskal-Wallis test was used to compare the groups.

A principal component analysis (PCA) was performed using the PCA program in R [18]. The significance of the correlation is determined by the correlation coefficient (*r*): strong—*r* = ± 0.700 to ± 1.00, medium—*r* = ± 0.300 to ± 0.699, weak—*r* = 0.00 to ± 0.299. The rate of change of individual biochemical parameters was quantified using the ratio of natural logarithms (LnRR), with LnRR (indicator) = ln (COPD / NO COPD value). Medians and interquartile range presented in figures were calculated using nontransformed data.

The survival curve was calculated by the Kaplan-Meier method and compared using the Log-rank test for univariate analysis (Statistica 10.0, StatSoft). Prognostic factors were analyzed by multivariate analysis using Cox’s proportional hazard model in a backward stepwise fashion to adjust for potential confounding factors. Overall survival (OS) was computed from the date of diagnosis to the date of death or the date of the last follow-up. Survival data were obtained until December 2019.

## 3. Results

### 3.1. Metabolic Features of Saliva in Lung Cancer Depending on the Presence/Absence of COPD

At the first stage, the biochemical composition of the saliva of patients with lung cancer was studied to identify metabolic changes (Table 2). It was found that the differences between the three groups are significant in the following indicators: pH, calcium concentration, the level of diene conjugates, sialic acids, and lactate dehydrogenase (LDH) activity (Table 2).

High values of the Kruskal-Wallis criterion were also noted for the Ca/P-ratio, magnesium concentration, catalase activity, alanine aminotransferase (ALT)/AST (aspartate aminotransferase)-ratio, and the level of Schiff bases (*p* < 0.10). All these indicators were used further for analysis by the PCA method (Figure 1). The first two dimensions of the analysis represent 34.25% of the total inertia of the dataset; this means that 34.25% of the total cloud variability is explained by the plane (Figure 1B). Estimating the correct number of axes for interpretation suggests limiting the analysis to describing the first five axes (principal components) (Figure 1B,D,F).

For the first principal component (PC1), high correlation coefficients (*p* = 0.001) were obtained only for LDH (*r* = 0.71), average strength for catalase (*r* = 0.61), magnesium (*r* = 0.53), calcium (*r* = 0.44) and pH (*r* = −0.47). For PC2, high correlation coefficients were not revealed, correlations of average strength were confirmed for catalase (*r* = 0.57), pH (*r* = 0.52), LDH (*r* = 0.41), Schiff bases (*r* = 0.35), calcium (*r* = −0.38) and sialic acids (*r* = −0.57). For PC3, correlations of average strength were established with Schiff bases (*r* = 0.68), calcium (*r* = 0.50), magnesium (*r* = 0.37), sialic acids (*r* = −0.35) and the AST/ALT-ratio (*r* = −0.33). For PC4 and PC5, only one strong correlation was revealed: with diene conjugates (*r* = 0.84) and the AST/ALT-ratio (*r* = 0.90), respectively (Figure 1). Thus, the given factor planes make it possible to divide the studied groups among themselves, but it should be borne in mind that the ellipses in the diagrams cover only 60% of the values that make the greatest contribution (Figure 1B,D,F). Thus, the most significant indicators affecting the composition of saliva in COPD are the activity of enzymes (LDH, catalase) (Figure 1B), then the level of lipid peroxidation products (diene conjugates, Schiff bases) (Figure 1D) and the value of AST/ALT-ratio are significant (Figure 1F).

### 3.2. Influence of Histological Type of Lung Cancer on Metabolic Indicators of Saliva in Patients with COPD

Since the group of patients with lung cancer is heterogeneous in its composition, at the next stage, each of the groups under consideration was divided into subgroups in accordance with the histological type of lung cancer (Table 3).

It was found that some of the indicators change unidirectionally for different histological types of lung cancer (Table 3). Regardless of the histological type of lung cancer, the pH increases in the presence of COPD. The maximum increase in pH was observed in COPD II. For adenocarcinoma, there is the smallest increase in pH (+2.0%), the largest for neuroendocrine lung cancer (+6.5%). For squamous cell lung cancer, the pH increases by 4.8%; it is for this group that the difference between the subgroups is statistically significant (*p* = 0.0106). The concentration of calcium and magnesium decreases in the presence of COPD. The decrease in calcium concentration is more pronounced for squamous cell carcinoma (−35.4%, *p* = 0.0053), while magnesium for adenocarcinoma (−23.4%, *p* = 0.0658). For diene conjugates and Schiff bases, a pronounced minimum is observed in COPD I, and then a sharp increase in their content. For LDH, on the contrary, there is a different nature of the change in activity depending on the histological type of lung cancer. Thus, for adenocarcinoma, LDH activity decreases (−5.4% for groups NO COPD vs. COPD I; −51.7% for COPD I vs. COPD II). For squamous cell and neuroendocrine cancers, LDH activity first increases (+1.0 and +18.1%, respectively) and then decreases (−34.0 and −1.5%, respectively). Lactate levels increase for adenocarcinoma and squamous cell carcinoma (+45.8 and +85.7%, respectively) but decrease for neuroendocrine lung cancer (−52.9%). The α-amylase activity increases for adenocarcinoma and neuroendocrine cancer (+19.1 and +127.9%, respectively) but decreases for squamous cell lung cancer (−14.0%).

For a visual representation of how the indicators change while taking into account the histological type of lung cancer and the severity of COPD, diagrams of the intensity of changes are plotted in comparison with the corresponding groups without COPD (Figure 2). To plot the diagram, only those biochemical indicators of saliva were selected, the change in which is statistically significant according to the Kruskal-Wallis test (Table 3). As can be seen from Figure 2, for all indicators except LDH and α-amylase, a wide scatter of data is observed. Changes in these indicators against the background of COPD of varying severity most reliably show the differences between histological types of lung cancer.

### 3.3. Biochemical Markers of Saliva in Patients with Lung Cancer, Noncancerous Pathologies of the Lung and Control Group Depending on the Presence/Absence of COPD

The differences between the groups with and without COPD are significant even within the group of patients with lung cancer; nevertheless, at the next stage of the study, the results obtained were compared with the control group and the comparison group. Table 4 shows the values of the indicators for each of the five groups and the corresponding value of the Kruskal-Wallis test. It was shown that the number of indicators by which the studied groups differ is significantly greater than for differences within the lung cancer group (Table 2).

The indicators, which turned out to be statistically significant according to the Kruskal-Wallis test (Table 4), were used for the analysis by the PCA method (Figure 3). It is shown that the first two dimensions of the analysis express 25.23% of the total inertia of the data set; therefore, for correct interpretation of the data, it is necessary to limit the analysis to the description of the first four axes (Figure 3).

For PC1, strong correlations were found only with protein content (*r* = 0.71), correlations of average strength with the activity of the enzymes LDH (*r* = 0.64), catalase (*r* = 0.58), GGT (*r* = 0.52) and ALP (*r* = 0.42), as well as with the content of α-amino acids (*r* = 0.49) and sialic acids (*r* = 0.31). For the other axes, high correlation coefficients were not found. For PC2, correlations of average strength were found for pH (*r* = 0.52), LDH enzymes (*r* = 0.36) and catalase (*r* = 0.44), diene conjugates (*r* = 0.33), and Schiff bases (*r* = 0.43), negative correlations were noted for protein metabolites: imidazole compounds (*r* = −0.41) and sialic acids (*r* = −0.45). For PC3, correlations of medium strength with imidazole compounds (*r* = 0.51) remain, correlations with diene conjugates (*r* = 0.56) and Schiff bases (*r* = 0.46) become more significant, and a correlation with the level of middle molecular toxins MM 280/254 (*r* = 0.37) is added. For PC4, the indicators of mineral metabolism are significant: pH (*r* = 0.34), calcium (*r* = 0.41), Ca/P-ratio (*r* = 0.56), as well as the level of α-amino acids (*r* = 0.32), diene conjugates (*r* = −0.35) and AST/ALT-ratio (*r* = −0.33).

As seen in Figure 3, the horizontal axis divides the study groups into a control group and a comparison group without COPD (above the axis), and patients with lung cancer, regardless of the presence/absence of COPD (below the axis) and a comparison group with COPD (below the axis). At the same time, the differences between the control group, the comparison group without COPD, and patients with lung cancer without COPD are less pronounced. In the diagram, these groups are located to the right of the vertical axis, while both groups of patients with COPD (lung cancer and the comparison group) are located to the left of the vertical axis (Figure 3B). The division into the listed groups provides differences in the content of protein metabolites (protein, α-amino acids, sialic acids and imidazole compounds) and enzymes (LDH, catalase, GGT, ALP and AST/ALT-ratio) in saliva. According to the severity of intoxication (indicators of lipid peroxidation and the level of medium molecular weight toxins), a control group can be distinguished both relative to the vertical and relative to the horizontal axes (Figure 3D).

### 3.4. Predictive Value of Saliva Biochemical Parameters Taking into Account the Presence/Absence of COPD

At the first stage of the study, we compared the overall survival rates (OS) of patients with lung cancer, as well as patients with concomitant pathology: lung cancer and COPD I and II (Figure 4A,B).

It was shown that, without taking into account the severity of COPD, the differences between OS indicators are insignificant HR = 1.17, 95% CI 0.79–1.74, *p* = 0.34185 (Figure 4A); in the case of combined pathology, the indicators of 1-year-old (63.9 vs. 60.8%), 3-year (32.3 vs. 31.1%) and 5-year survival rates (20.2 vs. 12.6%) decrease slightly. The median OS was 18.1 and 17.4 months, respectively. Taking into account the severity of COPD, it was found that the differences between patients with only lung cancer and those with COPD I are not significantly expressed HR = 0.86, 95% CI 0.56–1.32, while the differences between the groups of patients with lung cancer and COPD II statistically significant HR = 2.52, 95% CI 1.22–5.13, *p* = 0.05250 (Figure 4B). Differences between groups with COPD I and COPD II are also significant; HR = 2.94, 95% CI 1.37–6.20, *p* = 0.03878 (Figure 4B). It was found that the 1-year survival rates for patients with lung cancer, as well as patients with a combination of lung cancer with COPD I and COPD II, were 63.9, 62.3 and 56.3%, while the 3-year survival rates were 32.3, 38 and 13.7% and 5-year-old were 20.2, 19.7 and 13.7%, respectively (Figure 4B). The median of OS for the group of patients with COPD I increased slightly and amounted to 20.5 months, while against the background of COPD II, it decreased to 16.2 months.

For groups of lung cancer patients without COPD and with COPD (COPD I, II), the prognostic value of saliva biochemical indicators was determined by the method of multivariate regression. For the group of patients with lung cancer without COPD, such indicators were the level of imidazole compounds (ICs) and the activity of salivary LDH (Figure 5A).

It has been shown that the level of ICs less than 0.478 mmol/L and the LDH activity more than 1248 U/L are prognostically favorable signs. The selected values correspond to the median for LDH and the upper value of the interquartile range for ICs (Table 2). When combining indicators ICs < 0.478 and LDH >1248, a group of patients with a favorable prognosis can be distinguished (HR = 1.56, 95% CI 0.40–6.07, *p* = 0.03891), for which the median of OS is 11.8 months higher than for the group with a poor prognosis (Figure 5A). The differences between the other groups are not statistically significant.

For the group of patients with COPD of varying severity, the indices were divided: for COPD I, only the ICs level is significant, while for COPD II, the LDH activity and pH are significant (Figure 5B,C). For COPD I, the prognostically favorable ICs level is less than 0.182 mmol/L (HR = 1.74, 95% CI 0.71–4.21 and HR = 1.87, 95% CI 0.65–5.28, *p* = 0.07270, Figure 5B), while the median of OS with a favorable prognosis reaches 34.5 months. For COPD II, pH values less than 6.74 and LDH activity less than 1006 U/L are predictively favorable. In this case, we identified a group of patients with a poor prognosis HR = 2.17, 95% CI 0.43–10.79, *p* = 0.08977 (pH > 6.74 and LDH > 1006 U/L), for which the median of OS was 7.3 months; this is 2.2 times lower than the average for the group of patients with COPD II (Figure 5C). Some groups contain a small number of patients; therefore, the differences are not statistically significant and can be considered as preliminary data requiring verification on more representative samples of patients.

## 4. Discussion

It has long been assumed that there is a link between persistent chronic inflammation and lung cancer. Complex inflammatory processes involving many types of immune cells can cause tissue damage, leading to the development of COPD and lung cancer [19]. Over the past decade, there has been an increase in interest in the identification of biomarkers, in particular in COPD [20]. Some studies have observed a reduction in the risk of lung cancer in patients using inhaled corticosteroids [21]. Consequently, the likelihood of lung cancer due to COPD is increased, especially in cases where chronic inflammation is present along with the action of carcinogenic compounds of tobacco.

We have shown that in the group of patients with lung cancer, 43.1% have COPD of varying severity as a concomitant pathology, while in the comparison group, this percentage is significantly lower than 16.1%. In accordance with the histological type of lung cancer, COPD was detected in 40.2% of patients with adenocarcinoma, 48.9% with SCC, and 39.7% with neuroendocrine lung cancer. It is known that inflammatory processes in squamous cell lung cancer are more pronounced, which may explain the higher proportion of patients with concomitant COPD. Our data, however, somewhat contradict the literature, according to which 98% of lung cancer cases in patients with COPD are non-small cell lung cancer [22]. In our case, the share of patients with COPD accounts for 84% of non-small cell lung cancer, but this may be due to the characteristics of the sample.

The saliva composition of patients with lung cancer and COPD of varying severity has its own characteristics. Thus, the most pronounced differences between the groups in terms of electrolyte balance (pH, calcium and magnesium) and enzyme activity (catalase, LDH). The same parameters remain important when considering the histological type of lung cancer. It is known that hypercalcemia is relatively common in patients with lung cancer [23,24]; however, we have shown that, against the background of combined pathology of lung cancer and COPD, the level of calcium in saliva decreases. Some studies have shown that oxidative stress, defined as an imbalance between antioxidant defenses and the production of reactive oxygen species, is exacerbated in lung cancer in combination with COPD and can cause cellular dysfunction, DNA damage, and protein and lipid peroxidation [25,26,27,28]. This fact is confirmed by the high correlation coefficients for diene conjugates and Schiff bases (Figure 1C), as well as by a decrease in the activity of the main antioxidant enzyme in saliva, catalase [29]. Sialic acids play an important role in the differentiation of the studied groups; against the background of COPD, their content significantly decreases, which is consistent with our earlier data [30]. The literature describes the use of 13 glycans for differentiating lung cancer, COPD and their comorbidity from control and lung cancer from COPD [31]. In the cited work, it was shown that changes in the subclasses of N-glycans, in particular sialylation, can carry interesting diagnostic information.

It should be noted that when taking into account the histological type of tumor, two additional parameters appear, the activity of α-amylase and the level of lactate in saliva. Earlier, we showed that the activity of salivary α-amylase significantly increases against the background of lung adenocarcinoma, which coincides with the literature data, and the level of salivary amylase is increasing [32]. For neuroendocrine lung cancer, we have shown an increase in salivary amylase activity for the first time. In this case, the revealed patterns persist against the background of COPD. Hyperlactatemia can be considered as part of the stress response, which includes an increase in metabolic rate, activation of the sympathetic nervous system, accelerated glycolysis, and modified bioenergy reserves [33]. An increase in lactate levels may also be due to a corresponding decrease in LDH activity in the presence of COPD.

Despite the differences within the lung cancer group, when trying to compare the metabolic saliva profile of this group with the control group and the comparison group, it was likely that the division of groups with COPD of different severity would not occur. Nevertheless, it has been shown that, first, groups of patients without COPD are distinguished (control group and comparison group), and indicators of protein metabolism (protein, α-amino acids, sialic acids and imidazole compounds), as well as metabolic enzymes. All this indicates significant metabolic changes that are observed in lung cancer, including in combination with COPD. Against the background of COPD, there is a more pronounced change in the content of individual components in saliva (Table 2). Thus, it has been shown that it is possible to monitor metabolic changes in lung cancer by saliva, while the presence of COPD as a concomitant disease does not radically change the metabolic profile but increases the range of changes in the corresponding biochemical parameters.

We have found that some of the studied biochemical parameters may have a prognostic value in lung cancer. Therefore, for the group of patients without COPD, these indicators include imidazole compounds and LDH, with COPD I—only imidazole compounds, with COPD II—pH and LDH. In the first two cases, groups with a favorable prognosis were identified, for which the median overall survival was significantly higher (26.3 vs. 18.1 months and 34.5 vs. 20.5 months, respectively). In the case of COPD II, we managed to identify a group with a poor prognosis: the median was 7.3 vs. 16.2 months. Such prognostic results using saliva were obtained for the first time. Earlier, a comparison was made of the mortality rate in the groups of patients with COPD and patients with combined pathology of COPD and NSCLC [34,35]. NSCLC has been shown to have a higher mortality rate in patients with mild to moderate COPD (HR = 2.62, 95% CI 1.47–4.68), while in patients with severe/very severe COPD, mortality rates in lung cancer increased insignificantly (HR = 1.22, 95% CI 0.71–2.08) [36]. In general, according to the literature, COPD is associated with a poor prognosis; for example, 3-year mortality among patients with COPD in the GOLD 4 group was 25.8% compared to 4% in the group of patients with COPD in the GOLD 1 group [37].

In our study, only four cases of severe COPD were identified, so they were not taken into account in statistical processing. We obtained data according to which the survival rates of patients with mild COPD were not only not worse but also even better than those of the group with lung cancer without COPD. One of the probable reasons is the uneven distribution of different histological types of lung cancer in groups. Thus, the group without COPD includes 50.7% adenocarcinomas, 30.9% squamous cell and 18.4% neuroendocrine cancers, while the corresponding values for COPD I group are 50.0, 35.1 and 14.9%, for COPD II group—35.3, 45.1 and 19.6%. The median survival for adenocarcinoma calculated for the study group is 23.2 months, for squamous cell carcinoma 15.2 months, for neuroendocrine carcinoma—11.1 months. Thus, HR = 1.81 (95% CI 1.10–2.95) was for SSC and HR = 2.69 (95% CI 1.23–5.77) for neuroendocrine cancer (*p* < 0.00001). Therefore, it can be assumed that the prevalence of patients with adenocarcinoma in the COPD I group improves the survival rates for this group, while in the COPD II group, the proportion of patients with squamous cell and neuroendocrine cancers with a less favorable prognosis significantly increases. It is known that associations between COPD and worse survival outcomes were stronger in squamous cell carcinoma [38]. The group of patients with COPD II accounts for the maximum proportion of cancer with central growth (42.6%), while for the COPD I group and without COPD, the corresponding values were 33.8 and 27.3%. The share of peripheral cancer was 51.5, 62.5 and 69.2% for the groups with COPD II, COPD I and without COPD. It is also one of the factors in the less favorable prognosis for lung cancer combined with COPD II. By the nature of the treatment carried out, the study groups are comparable. Complete radical treatment was performed in 22.1, 28.1, and 26.9%, combined—in 19.1, 28.8 and 28.0%, palliative—in 39.7, 32.3 and 34.8% of patients with COPD II, COPD I and without COPD. For the group of patients with COPD II, the proportion of patients who are not indicated for special treatment methods (19.1%) is higher; for the groups with COPD I and without COPD, the corresponding values were 10.8 and 10.2%. These data show that the group with lung cancer in combination with COPD I includes a larger number of patients who are indicated for radical and combined treatment, and therefore higher survival rates are quite justified. Nevertheless, we have shown that when comparing the metabolic parameters of saliva for the group of patients with lung cancer and COPD I, the values are closer to the control than those for the group of patients with lung cancer without COPD. This may be due to the formation of compensatory mechanisms in the early stages of COPD, which requires more detailed research in the course of further work.

The limitations of the study are associated with the absence of patients with severe COPD (GOLD III and GOLD IV) and with the absence of patients with COPD without other pulmonary pathologies (control group). The small sample size of patients with lung cancer and COPD II limits the possibility of dividing this group into subgroups and reduces the statistical significance of the results.

## 5. Conclusions

Thus, saliva adequately reflects metabolic characteristics in lung cancer, as well as lung cancer in combination with COPD of varying severity. It has been shown that the presence of COPD as a concomitant pathology does not fundamentally change the metabolic profile of saliva but increases the range of changes in the corresponding biochemical parameters. A complex of biochemical parameters of saliva, which have prognostic value in lung cancer, was revealed. It should be noted that in the presence of COPD, the list of parameters differs from that without COPD, which must be taken into account when planning treatment tactics and assessing the prognosis of lung cancer. In general, for patients with lung cancer in combination with mild COPD, the prognosis is more favorable than without COPD, which is due to compensatory mechanisms. The revealed biochemical parameters (catalase, imidazole compounds, sialic acids, LDH) can be used to monitor patients from risk groups, for example, groups of patients with initial stages of COPD, in particular for the timely diagnosis of lung cancer.

## Figures and Tables

**Figure 1 diagnostics-10-01095-f001:**
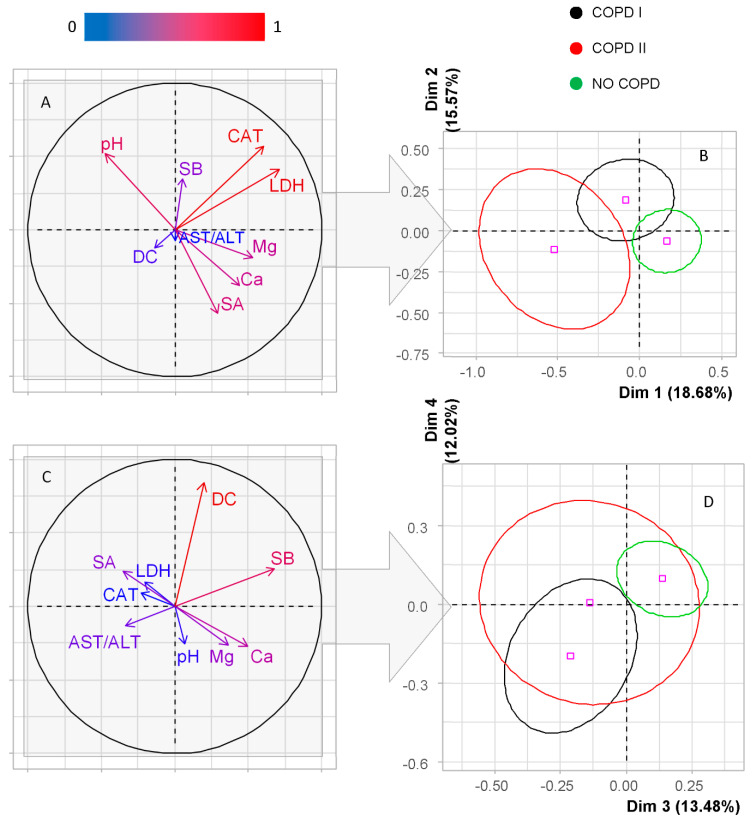
Principal component analysis (PCA)—Factor plane (**B**,**D**,**F**) and the corresponding correlation circle (**A**,**C**,**E**) in the first five dimensions for three groups (lung cancer without COPD, lung cancer with COPD I, lung cancer with COPD II). The color of the arrows on the correlation circle changes from blue (weak correlation) to red (strong correlation), as shown on the color bar. The orientation of the arrows characterizes positive and negative correlations (for the first principal component, we analyze the location of the arrows relative to the vertical axis, for the second principal component—relative to the horizontal axis). LDH—lactate dehydrogenase; CAT—catalase; SB—Schiff bases; DC—diene conjugates; SA—sialic acids; Ca—calcium; Mg—magnesium; AST/ALT—aspartate aminotransferase to alanine aminotransferase ratio.

**Figure 2 diagnostics-10-01095-f002:**
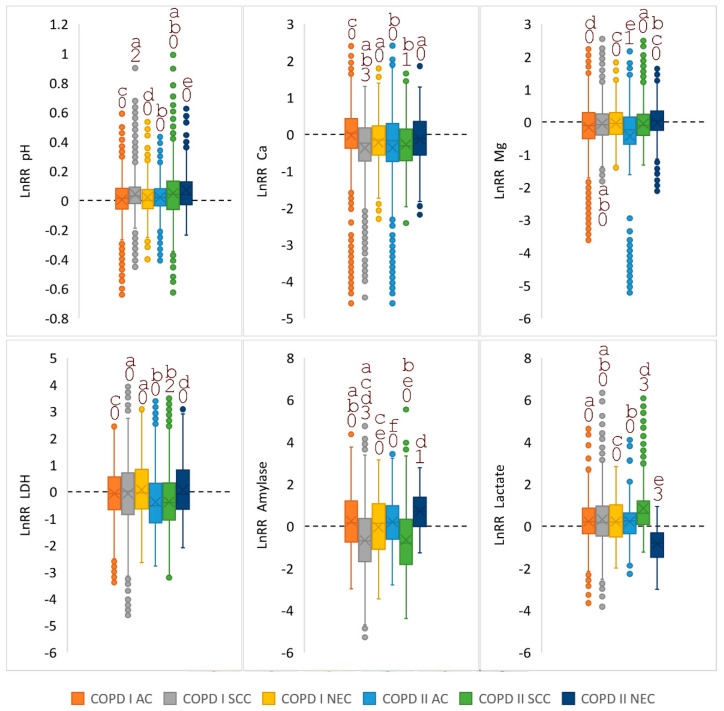
The intensity of the difference (LnRR = Ln (COPD/NO COPD)) in saliva biochemical parameters for different histological types of lung cancer, depending on the presence/absence of COPD. The rectangles show the mean (line) and median (cross); individual points show values outside the interquartile range. The same letters (a, b, c, d, e, f) denote groups, the differences between which were not revealed (α = 0.05). The dotted line represents the zero level (LnRR = 0). Significance of difference from 0 was checked using the t-test of one sample and indicated by numbers (“0”: not significant; “1”: *p* < 0.05; “2”: *p* < 0.01; “3”: *p* < 0.001). AC—adenocarcinoma; SCC—squamous cell carcinoma; NEC—neuroendocrine lung cancer.

**Figure 3 diagnostics-10-01095-f003:**
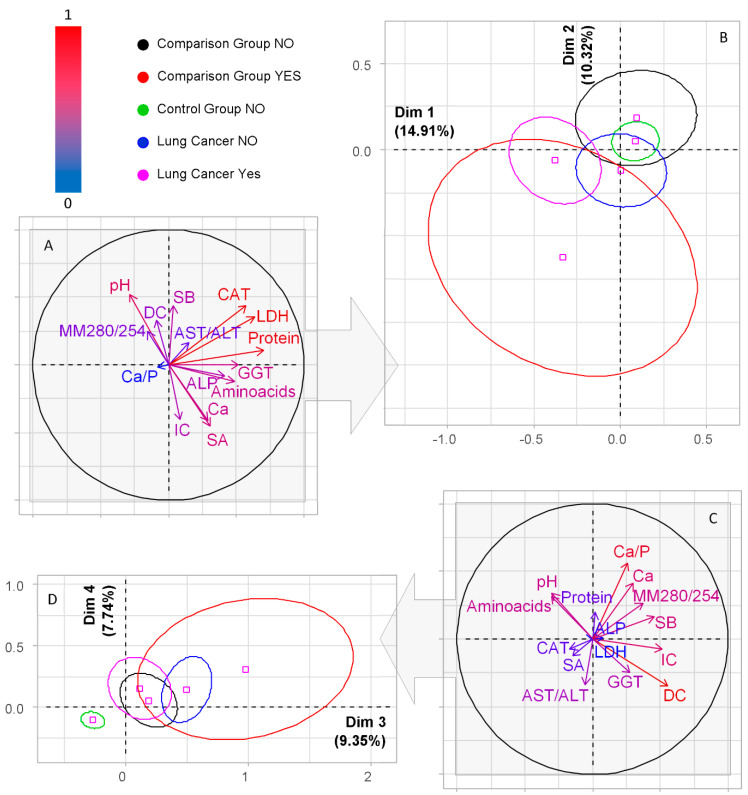
PCA—Factor plane (**B**,**D**) and the corresponding correlation circle (**A**,**C**) in the first four dimensions for five groups (control, lung cancer without COPD, lung cancer with COPD, comparison group without COPD, comparison group with COPD). The color of the arrows on the correlation circle changes from blue (weak correlation) to red (strong correlation), as shown on the color bar. The orientation of the arrows characterizes positive and negative correlations (for the first principal component, we analyze the location of the arrows relative to the vertical axis, for the second principal component-relative to the horizontal axis). LDH—lactate dehydrogenase; CAT—catalase; SB—Schiff bases; DC—diene conjugates; SA—sialic acids; Ca—calcium; Mg—magnesium; Ca/P—calcium to phosphorus ratio; IC—imidazole compounds; GGT—gamma glutamyltransferase; MM 280/254—middle molecular toxins; AST/ALT—aspartate aminotransferase to alanine aminotransferase ratio; ALP—alkaline phosphatase.

**Figure 4 diagnostics-10-01095-f004:**
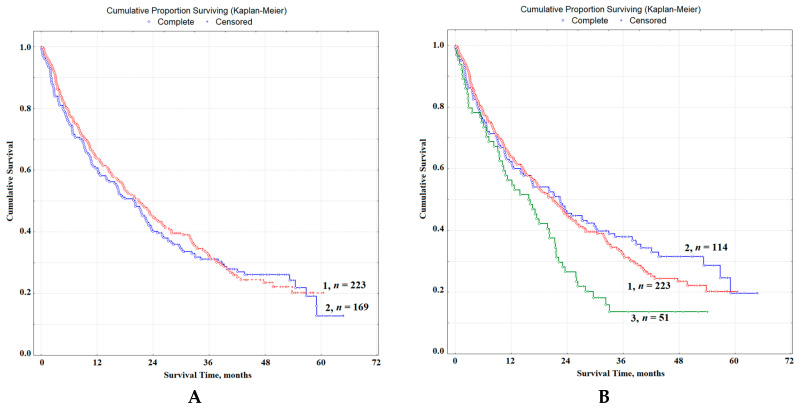
Overall survival of patients with lung cancer: (**A**) without COPD (curve 1) and in the presence of COPD (curve 2); (**B**) without COPD (curve 1), lung cancer with COPD I (curve 2) and lung cancer with COPD II (curve 3). *n* is the number of patients in each group.

**Figure 5 diagnostics-10-01095-f005:**
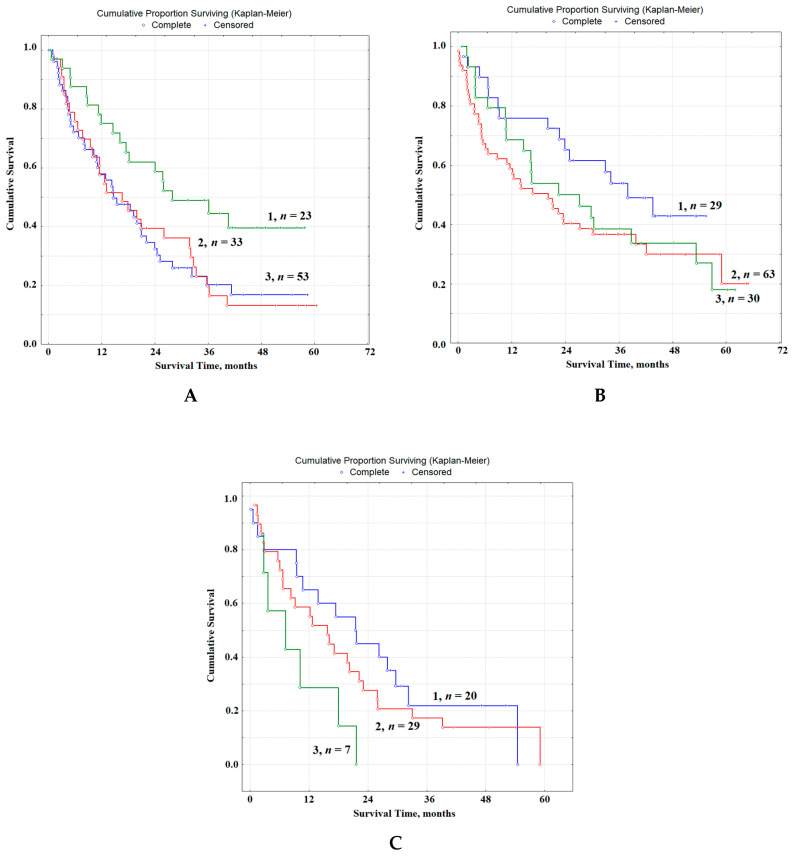
Multivariate regression analysis of overall survival: (**A**) ICs < 0.478 mmol/L and LDH > 1248 U/L (curve 1, favorable prognosis, median 26.3 months), ICs > 0.478 mmol/L and LDH < 1248 U/L (curve 2, poor prognosis, median 16.9 months), other combinations (curve 3, median 14.5 months); (**B**) ICs < 0.182 mmol/L (curve 1, median 34.5 months), 0.182 < ICs < 0.455 mmol/L (curve 2, median 14.3 months), ICs > 0.455 mmol/L (curve 3, median 16.7 months); (**C**) pH < 6.74 and LDH < 1006 U/L (curve 1, favorable prognosis, median 21.9 months), pH > 6.74 and LDH > 1006 U/L (curve 3, poor prognosis, median 7.3 months), other combinations (curve 2, median 16.0 months). *n* is the number of patients in each group.

**Table 1 diagnostics-10-01095-t001:** The structure of the study group.

Feature			Lung Cancer, *n* (%)			Non-Malignant Lung Diseases, *n* = 168
			ADC, *n* = 189	SCC, *n* = 135	NEC, *n* = 68	
Age, years			61.0 (56.0; 65.0)	59.0 (55.0; 66.5)	55.0 (52.0; 60.0)	55.0 (45.5; 60.5)
Gender						
	Male		129 (68.3)	128 (94.8)	50 (73.5)	98 (58.3)
	Female		60 (31.7)	7 (5.2)	18 (26.5)	70 (41.7)
Stage						
	St IA		16 (8.5)	3 (2.2)	5 (7.4)	-
	St IB		52 (27.5)	28 (20.7)	10 (14.7)	-
	St IIA+B		23 (12.2)	19 (14.1)	6 (8.8)	-
	St IIIA		25 (13.2)	34 (25.2)	10 (14.7)	-
	St IIIB		17 (9.0)	24 (17.8)	17 (25.0)	-
	St IV		56 (29.6)	27 (20.0)	20 (29.4)	-
COPD						
	No		113 (59.8)	69 (51.1)	41 (60.3)	141 (83.9)
	Yes	GOLD I	57 (30.2)	40 (29.6)	17 (25.0)	27 (16.1)
		GOLD II	18 (9.5)	23 (17.1)	10 (14.7)	-
		GOLD III	1 (0.5)	3 (2.2)	-	-

**Table 2 diagnostics-10-01095-t002:** Biochemical markers of saliva in patients with lung cancer, depending on the presence/absence and severity of cancer, chronic obstructive pulmonary disease (COPD).

Indicators		Lung Cancer (LC)	LC + COPD I	LC + COPD II		Kruskal-Wallis Test (*H*, *p*)
Flow rate, mL/min		0.85 (0.72; 0.99)	0.81 (0.74; 0.94)	0.78 (0.64; 0.97)	↓↓	0.8541, 0.7596
Electrolytes						
	pH	6.46 (6.20; 6.68)	6.57 (6.32; 6.85)	6.63 (6.28; 7.03)	↑↑	11.20, 0.0037 *
	Calcium (Ca), mmol/L	1.48 (1.07; 1.92)	1.41 (1.02; 1.80)	1.19 (0.83; 1.51)	↓↓	8.900, 0.0117 *
	Phosphorus (P), mmol/L	4.61 (3.37; 5.78)	4.00 (2.97; 5.32)	4.55 (3.37; 6.01)	↓↑	3.979, 0.1368
	Ca/P-ratio, c.u.	0.32 (0.23; 0.47)	0.33 (0.23; 0.51)	0.28 (0.17; 0.39)	=↑	4.737, 0.0936 **
	Sodium (Na), mmol/L	9.0 (5.4; 13.5)	9.6 (6.1; 13.8)	8.3 (5.9; 16.5)	↑↓	0.9659, 0.6170
	Potassium (K), mmol/L	12.9 (9.4; 16.0)	13.0 (8.5; 15.9)	12.8 (9.3; 16.6)	==	0.3176, 0.8532
	Na/K-ratio, c.u.	0.73 (0.45; 1.23)	0.87 (0.48; 1.31)	0.76 (0.46; 1.19)	↑↓	1.058, 0.5892
	Chlorides, mmol/L	28.3 (22.3; 35.7)	27.5 (21.0; 37.2)	29.8 (21.0; 36.4)	↓↑	0.1096, 0.9467
	Magnesium, mmol/L	0.311 (0.248; 0.381)	0.286 (0.220; 0.361)	0.279 (0.213; 0.346)	↓↓	5.981, 0.0503 **
Protein Metabolism						
	Protein, g/L	0.64 (0.33; 1.10)	0.65 (0.36; 0.94)	0.64 (0.40; 0.96)	==	0.7507, 0.6870
	Albumin, g/L	0.313 (0.154; 0.475)	0.286 (0.153; 0.462)	0.311 (0.178; 0.485)	↓↑	0.9102, 0.6344
	Urea, mmol/L	8.06 (5.34; 11.60)	7.78 (6.09; 11.64)	7.34 (4.49; 11.60)	↓↓	0.4845, 0.7849
	Uric acid, µmol/L	90.4 (45.0; 181.0)	76.7 (34.7; 133.7)	100.0 (24.8; 195.2)	↓↑	4.346, 0.1138
	Sialic acids, mmol/L	0.189 (0.110; 0.287)	0.134 (0.079; 0.238)	0.180 (0.093; 0.314)	↓↑	6.161, 0.0459 *
	Seromucoids, c.u.	0.104 (0.056; 0.159)	0.097 (0.060; 0.149)	0.081 (0.039; 0.134)	↓↓	3.464, 0.1770
	α-amino acids, µmol/L	4.16 (3.87; 4.66)	4.12 (3.86; 4.53)	4.16 (3.93; 4.49)	↓↑	0.5360, 0.7649
	Imidazole compounds, mmol/L	0.296 (0.197; 0.478)	0.319 (0.182; 0.455)	0.341 (0.266; 0.531)	↑↑	1.995, 0.3688
	Lactate, µmol/L	2.32 (1.59; 3.50)	2.75 (1.71; 3.96)	3.12 (1.85; 4.14)	↑↑	2.702, 0.2590
Enzymes						
	ALT, U/L	4.00 (2.69; 6.00)	3.77 (2.62; 4.96)	4.15 (2.69; 6.54)	↓↑	2.651, 0.2657
	AST, U/L	5.58 (3.33; 8.08)	4.96 (3.25; 6.99)	4.58 (3.04; 6.75)	↓↓	3.854, 0.1456
	AST/ALT, c.u.	1.29 (1.01; 1.70)	1.32 (1.03; 1.66)	1.07 (0.86; 1.55)	↑↓	5.026, 0.0810 **
	ALP, U/L	76.06 (49.98; 117.34)	63.02 (47.81; 97.79)	73.88 (52.15; 130.38)	↓↑	2.284, 0.3192
	LDH, U/L	1248.5(604.3; 1907.0)	1165.0(605.2; 1715.0)	764.4(481.0; 1206.3)	↓↓	9.210, 0.0100 *
	GGT, U/L	22.3 (18.5; 26.1)	20.6 (17.5; 25.5)	21.1 (18.5; 23.1)	↓↑	3.603, 0.1651
	α-amylase, U/L	317.4 (198.6; 655.7)	287.6 (99.9; 600.9)	303.3 (139.7; 680.2)	↓↑	1.529, 0.4656
	Catalase, ncat/mL	2.66 (2.04; 4.36)	2.78 (2.13; 4.13)	2.41 (1.73; 3.17)	↑↓	5.986, 0.0501 **
	SOD, c.u.	60.5 (26.3; 121.1)	61.8 (34.2; 110.5)	65.8 (39.5; 92.1)	↑↑	0.1997, 0.9050
Lipoperoxidation products and endogenous intoxication rates						
	Diene conjugates, c.u.	4.01 (3.83; 4.20)	3.91 (3.75; 4.13)	3.97 (3.79; 4.15)		6.136, 0.0465 *
	Triene conjugates, c.u.	0.892 (0.775; 1.004)	0.879 (0.792; 0.967)	0.907 (0.793; 1.009)	↓↑	0.7660, 0.6818
	Schiff bases, c.u.	0.561 (0.494; 0.675)	0.529 (0.474; 0.638)	0.556 (0.509; 0.652)	↓↑	5.054, 0.0799 **
	MDA, µmol/L	7.14 (5.81; 9.32)	6.67 (5.60; 9.27)	8.46 (6.15; 10.17)	↓↑	4.258, 0.1189
	MM 280/254 nm	0.897 (0.806; 1.002)	0.875 (0.794; 0.989)	0.906 (0.832; 1.055)	↓↑	1.878, 0.3914

*—differences between 3 groups are statistically significant, *p* < 0.05; **—differences between 3 groups are statistically significant, *p* < 0.10; the first arrow shows the change in the parameter when switching from the LC group to the LC + COPDI, the second—when changing from the LC + COPDI to LC + COPD II; ↑—the indicator value increases; ↓—the indicator value decreases; =—the indicator value does not change; SOD—superoxide dismutase; MDA—malondialdehyde; MM 280/254—middle molecular toxins; AST—aspartate aminotransferase; ALT—alanine aminotransferase; LDH—lactate dehydrogenase; GGT—gamma glutamyltransferase; ALP—alkaline phosphatase.

**Table 3 diagnostics-10-01095-t003:** Metabolic profile of saliva in patients with lung cancer of various histological types depending on the presence/absence of COPD.

Indicator	HT	NO COPD	COPD I	COPD II	Kruskal-Wallis Test (*H*, *p*)
pH	ADC	6.47 (6.22; 6.72)	6.57 (6.32; 6.85)	6.60 (6.29; 7.00)	2.508, 0.2853
	SCC	6.45 (6.23; 6.64)	6.61 (6.43; 6.90)	6.76 (6.13; 7.15)	9.085, 0.0106 *
	NEC	6.33 (6.03; 6.50)	6.43 (6.14; 6.60)	6.74 (6.28; 6.81)	2.706, 0.2584
Calcium, mmol/L	ADC	1.40 (1.03; 1.88)	1.60 (1.11; 1.84)	1.24 (0.77; 1.51)	2.239, 0.3264
	SCC	1.58 (1.14; 2.08)	1.27 (0.85; 1.68)	1.02 (0.83; 1.47)	10.49, 0.0053 *
	NEC	1.44 (1.28; 1.80)	1.25 (1.03; 1.57)	1.24 (0.98; 1.55)	1.948, 0.3776
Magnesium, mmol/L	ADC	0.320 (0.259; 0.406)	0.285 (0.216; 0.389)	0.245 (0.161; 0.343)	5.441, 0.0658 **
	SCC	0.302 (0.254; 0.378)	0.292 (0.228; 0.336)	0.277 (0.213; 0.349)	2.117, 0.3471
	NEC	0.299 (0.230; 0.342)	0.298 (0.219; 0.328)	0.287 (0.275; 0.398)	0.3195, 0.8524
AST/ALT, c.u.	ADC	1.38 (1.01; 1.73)	1.30 (1.08; 1.81)	1.03 (0.76; 1.28)	4.566, 0.1020
	SCC	1.17 (0.92; 1.56)	1.34 (1.01; 1.60)	1.05 (0.94; 1.58)	1.107, 0.5750
	NEC	1.31 (1.08; 1.69)	1.34 (0.99; 1.88)	1.32 (0.97; 1.54)	0.3129, 0.8552
LDH, U/L	ADC	1253.0 (655.1; 1842.0)	1185.0 (665.7; 1647.0)	572.3 (497.9; 1095.0)	5.428, 0.0663 **
	SCC	1193.0 (566.0; 1893.0)	1205.5 (492.4; 1918.5)	795.8 (437.7; 1108.0)	3.955, 0.1384
	NEC	986.8 (446.5; 1940.0)	1165.0 (1007.0; 1616.0)	1148.0 (724.3; 1430.0)	0.2333, 0.8899
Catalase, ncat/mL	ADC	2.74 (2.12; 4.42)	2.81 (2.18; 4.08)	2.29 (1.81; 3.04)	3.506, 0.1733
	SCC	2.69 (2.00; 4.29)	2.65 (2.12; 3.48)	2.68 (1.90; 3.17)	1.522, 0.4673
	NEC	2.36 (2.04; 3.38)	3.48 (2.60; 4.33)	1.72 (1.65; 3.24)	3.137, 0.2083
Sialic acids, mmol/L	ADC	0.159 (0.098; 0.275)	0.125 (0.092; 0.226)	0.168 (0.079; 0.323)	2.103, 0.3493
	SCC	0.201 (0.125; 0.281)	0.150 (0.069; 0.269)	0.183 (0.098; 0.232)	3.468, 0.1766
	NEC	0.204 (0.101; 0.336)	0.128 (0.092; 0.323)	0.293 (0.092; 0.488)	1.103, 0.5761
Diene conjugates, c.u.	ADC	4.01 (3.83; 4.18)	3.90 (3.67; 4.17)	4.00 (3.79; 4.15)	3.541, 0.1702
	SCC	4.03 (3.85; 4.21)	3.92 (3.78; 4.07)	3.96 (3.78; 4.23)	3.957, 0.1383
	NEC	4.05 (3.80; 4.15)	3.99 (3.85; 4.16)	3.88 (3.84; 3.92)	1.135, 0.5669
Schiff bases, c.u.	ADC	0.559 (0.499; 0.671)	0.529 (0.478; 0.638)	0.551 (0.495; 0.665)	2.627, 0.2688
	SCC	0.573 (0.490; 0.669)	0.538 (0.476; 0.646)	0.556 (0.507; 0.627)	1.402, 0.4960
	NEC	0.557 (0.495; 0.641)	0.487 (0.417; 0.633)	0.570 (0.532; 0.662)	1.589, 0.4517
α-amylase, U/L	ADC	316.5 (178.7; 645.0)	323.3 (184.7; 929.3)	377.1 (266.5; 680.2)	0.4174, 0.8116
	SCC	270.7 (198.6; 857.0)	232.3 (56.4; 360.3)	232.8 (90.1; 284.7)	6.986, 0.0304 *
	NEC	349.6 (223.9; 458.8)	477.1 (74.8; 965.0)	796.8 (379.5; 1278.0)	2.126, 0.3454
Lactate, nmol/mL	ADC	2.14 (1.66; 3.12)	2.75 (1.59; 4.41)	3.12 (1.31; 3.43)	1.032, 0.5971
	SCC	2.23 (1.29; 3.36)	2.74 (1.77; 3.96)	4.14 (2.79; 4.74)	2.887, 0.2361
	NEC	3.76 (2.68; 6.44)	3.01 (2.28; 7.97)	1.77 (0.76; 1.85)	5.381, 0.0479 *

HT—histology type; ADC—adenocarcinoma; SCC—squamous cell carcinoma; NEC—neuroendocrine cancer; AST—aspartate aminotransferase; ALT—alanine aminotransferase; LDH—lactate dehydrogenase; *—differences between 3 groups are statistically significant, *p* < 0.05; **—differences between 3 groups are statistically significant, *p* < 0.10.

**Table 4 diagnostics-10-01095-t004:** Biochemical markers of saliva in patients with lung cancer, noncancerous pathologies of the lung and control group.

Indicator		Control Group		Lung Cancer	Comparison Group	Kruskal-Wallis Test (*H*, *p*)
Electrolytes						
	pH	6.50 (6.30; 6.72)	1	6.46 (6.20; 6.68)	6.49 (6.24; 6.80)	14.11, 0.0070 *
			2	6.59 (6.32; 6.89)	6.34 (6.08; 6.82)	
	Calcium, mmol/L	1.33 (1.05; 1.66)	1	1.48 (1.07; 1.92)	1.36 (1.02; 1.69)	13.75, 0.0081 *
			2	1.30 (0.97; 1.76)	1.44 (1.16; 2.35)	
	Phosphorus, mmol/L	4.53 (3.58; 5.85)	1	4.61 (3.37; 5.78)	4.74 (3.47; 5.64)	5.955, 0.2026
			2	4.13 (3.25; 5.53)	4.80 (3.17; 6.54)	
	Ca/P-ratio, c.u.	0.29 (0.22; 0.39)	1	0.32 (0.23; 0.47)	0.29 (0.20; 0.40)	11.34, 0.0230 *
			2	0.31 (0.21; 0.47)	0.34 (0.17; 0.53)	
	Sodium, mmol/L	8.4 (5.5; 12.4)	1	9.0 (5.4; 13.5)	7.8 (5.9; 10.7)	5.546, 0.2357
			2	9.3 (6.1; 15.0)	9.5 (4.8; 15.3)	
	Potassium, mmol/L	11.8 (9.3; 14.7)	1	12.9 (9.4; 16.0)	12.4 (9.4; 14.8)	6.576, 0.1601
			2	12.8 (8.9; 15.9)	9.7 (7.2; 13.9)	
	Na/K-ratio, c.u.	0.72 (0.49; 1.15)	1	0.73 (0.45; 1.23)	0.74 (0.47; 1.08)	3.033, 0.5524
			2	0.83 (0.48; 1.30)	0.90 (0.55; 1.49)	
	Chlorides, mmol/L	26.1 (21.2; 32.2)	1	28.3 (22.3; 35.7)	25.4 (20.7; 33.5)	9.383, 0.0522
			2	28.2 (21.0; 36.8)	23.3 (19.2; 34.3)	
	Mg, mmol/L	0.300 (0.246; 0.350)	1	0.311 (0.248; 0.381)	0.295 (0.241; 0.366)	6.389, 0.1719
			2	0.286 (0.220; 0.349)	0.313 (0.221; 0.400)	
Protein metabolism						
	Protein, g/L	0.80 (0.50; 1.23)	1	0.64 (0.33; 1.10)	0.69 (0.49; 1.11)	20.78, 0.0004 *
			2	0.65 (0.36; 0.95)	0.69 (0.54; 1.02)	
	Albumin, g/L	0.258 (0.171; 0.435)	1	0.313 (0.154; 0.475)	0.337 (0.178; 0.543)	7.562, 0.1090
			2	0.295 (0.162; 0.474)	0.304 (0.178; 0.498)	
	Urea, mmol/L	7.84 (5.40; 11.03)	1	8.06 (5.34; 11.60)	7.78 (5.27; 10.78)	4.087, 0.3944
			2	7.57 (5.81; 11.60)	6.12 (3.84; 9.06)	
	Uric acid, µmol/L	86.5 (28.2; 154.8)	1	90.4 (45.0; 181.0)	84.9 (30.4; 163.4)	6.472, 0.1666
			2	80.4 (34.7; 146.8)	85.0 (22.3; 176.6)	
	Sialic acids, mmol/L	0.195 (0.134; 0.299)	1	0.189 (0.110; 0.287)	0.165 (0.104; 0.238)	35.35, 0.0000 *
			2	0.146 (0.085; 0.262)	0.134 (0.079; 0.201)	
	Seromucoids, c.u.	0.090 (0.060; 0.130)	1	0.104 (0.056; 0.159)	0.105 (0.066; 0.149)	6.607, 0.1582
			2	0.092 (0.055; 0.141)	0.095 (0.046; 0.146)	
	α-amino acids, µmol/L	4.12 (3.83; 4.50)	1	4.16 (3.87; 4.66)	4.25 (4.00; 4.60)	13.05, 0.0110 *
			2	4.13 (3.87; 4.51)	4.02 (3.77; 4.50)	
	Imidazole compounds, mmol/L	0.281 (0.175; 0.379)	1	0.296 (0.197; 0.478)	0.326 (0.212; 0.448)	30.09, 0.0000 *
			2	0.326 (0.212; 0.478)	0.448 (0.197; 0.660)	
	Lactate, µmol/L	2.36 (1.61; 3.48)	1	2.32 (1.59; 3.50)	2.65 (1.65; 3.24)	3.357, 0.5000
			2	2.81 (1.77; 4.10)	1.54 (0.36; 4.21)	
Enzymes						
	ALT, U/L	3.62 (2.54; 4.92)	1	4.00 (2.69; 6.00)	3.96 (2.85; 6.31)	13.56, 0.0088 *
			2	3.88 (2.62; 5.38)	3.58 (2.54; 5.23)	
	AST, U/L	5.50 (3.67; 7.33)	1	5.58 (3.33; 8.08)	6.00 (4.17; 8.67)	12.22, 0.0158 *
			2	4.92 (3.17; 6.99)	4.29 (3.08; 6.75)	
	AST/ALT, c.u.	1.42 (1.13; 1.92)	1	1.29 (1.01; 1.70)	1.42 (1.08; 1.83)	23.45, 0.0001 *
			2	1.25 (0.97; 1.64)	1.25 (1.00; 1.76)	
	ALP, U/L	58.7 (41.3; 82.6)	1	76.1 (50.0; 117.3)	71.7 (52.2; 115.2)	38.57, 0.0000 *
			2	69.5 (47.8; 108.7)	69.5 (50.0; 117.3)	
	LDH, U/L	1127.5 (652.1; 1838.0)	1	1248.5 (604.3; 1907.0)	1291.0 (762.0; 1902.0)	10.17, 0.0377 *
			2	1060.0 (545.5; 1616.0)	883.2 (527.7; 1443.0)	
	GGT, U/L	20.3 (17.5; 24.0)	1	22.3 (18.5; 26.1)	21.6 (18.1; 25.4)	15.29, 0.0041 *
			2	20.8 (18.0; 25.1)	20.1 (16.2; 25.0)	
	α-amylase, U/L	201.6 (100.5; 404.4)	1	317.4 (198.6; 655.7)	316.0 (178.6; 468.4)	23.54, 0.0001 *
			2	287.6 (122.2; 618.1)	168.1 (27.9; 804.2)	
	Catalase, ncat/mL	4.32 (3.20; 5.57)	1	2.66 (2.04; 4.36)	3.28 (2.34; 5.06)	160.5, 0.0000 *
			2	2.70 (1.97; 3.62)	2.57 (1.76; 3.52)	
	SOD, c.u.	57.9 (34.2; 103.9)	1	60.5 (26.3; 121.1)	63.2 (42.1; 131.6)	2.971, 0.5627
			2	63.2 (39.5; 110.5)	63.2 (52.6; 84.2)	
Lipoperoxidation products and endogenous intoxication rates						
	Diene conjugates, c.u.	3.92 (3.78; 4.06)	1	4.01 (3.83; 4.20)	3.97 (3.76; 4.13)	21.50, 0.0003 *
			2	3.92 (3.77; 4.14)	4.04 (3.77; 4.25)	
	Triene conjugates, c.u.	0.870 (0.793; 0.944)	1	0.892 (0.775; 1.004)	0.901 (0.824; 1.010)	7.994, 0.0918
			2	0.890 (0.792; 0.981)	0.861 (0.792; 0.958)	
	Schiff bases, c.u.	0.528 (0.492; 0.565)	1	0.561 (0.494; 0.675)	0.555 (0.492; 0.686)	38.33, 0.0000 *
			2	0.541 (0.480; 0.640)	0.563 (0.470; 0.661)	
	MDA, µmol/L	6.84 (5.81; 8.38)	1	7.14 (5.81; 9.32)	7.39 (5.94; 9.44)	8.374, 0.0788
			2	7.35 (5.64; 9.91)	6.03 (5.13; 9.10)	
	MM 280/254 nm	0.847 (0.749; 0.948)	1	0.897 (0.806; 1.002)	0.892 (0.812; 0.970)	28.88, 0.0000 *
			2	0.894 (0.802; 1.020)	0.893 (0.776; 1.075)	

1—Groups without COPD, 2—groups with COPD; *—differences between the 5 groups are statistically significant (*p* < 0.05); SOD—superoxide dismutase; MDA—malondialdehyde; MM 280/254—middle molecular toxins; AST—aspartate aminotransferase; ALT—alanine aminotransferase; LDH—lactate dehydrogenase; GGT—gamma glutamyltransferase; ALP—alkaline phosphatase.

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
