# Peer review of "Salivary Metabolic Profile of Patients with Lung Cancer, Chronic Obstructive Pulmonary Disease of Varying Severity and Their Comorbidity: A Preliminary Study"

_diagnostics, 2020, doi:10.3390/diagnostics10121095_

Round 1

Reviewer 1 Report

The authors studied the features of the salivary biochemical composition in the patients with combined pathology of lung cancer and COPD of varying severity and concluded that saliva adequately reflects metabolic characteristics in lung cancer, as well as lung cancer in combination with COPD of varying severity.

The design of the study is well performed. The results are largely accompanied by numerous tables and figures that help the readers to follow the results.

The discussion is well written and relevant literature is included.

As tumor type was included as an important part of the study, why local pathologist was not inlcuded as a co-author?

The references are relevant.

Author Response

The authors of the article are grateful to the reviewers for valuable comments that made it possible to eliminate inaccuracies and make the article better.

Reviewer 1

As tumor type was included as an important part of the study, why local pathologist was not inlcuded as a co-author?

The data on the histological verification of the diagnosis were obtained from the patient's medical records; the consent of each participant in the experiment was obtained. We did not work directly with any of the pathologists.

Reviewer 2 Report

The authors present a large amount of data that is described in a very superficial and confused way. 

The authors use various statistical methods normally used in medicine (epidemiology), however, the way they describe the results suggests they have limited experience in the field. Alternatively, they have a vast experience and do not consider it necessary to provide detailed indications on the criteria used in their analysis. 

Figures have inappropriate legends. In some cases, they mix up the letter that indicates the various insets. 

The English form is very poor making it even more difficult to understand what they describe. 

The authors must keep in mind that scientific work must be readable and understandable: e.g. I am referring to the description of the statistical analysis that is 100% of the work. 

I cannot pin down specific point to be amended, corrected, or modified, as I usually do, because there are too many points to raise. As a general comment: 

Some explanations on the choice of a non-parametric method should be given. 

The description of the PCA factor planes is approximate. It is difficult to read the various factor planes. No indications of the criteria used to determine intensity, orientation, as well as color of the arrows, is reported. Reading the text, it seems there are incongruences. 

A comment on how the H factor has been used to analyze the data would be useful. 

In the legend of Fig 1 it is written "factor plane (b, e, f)" that is incorrect, and similar errors are in other figures 

Describing the correlation circle of Fig 3 the authors state that "as well as patients with lung cancer, regardless of the presence/absence of COPD" are above the horizontal axis, incorrect. 

Fig 4 reports understandable indications only for curve in inset a and b, for the other nothing is reported. Besides the number "n" reported at the end of any curve, and that I assume it corresponds to the number of cases, varies from tenth to hundreds, making any comparison dubious. A comment on size and position of the rectangles is necessary. 

Fig 2 nothing is said to facilitate the reading of the various graph 

These are only a few examples. 

Overall, due to the confusing description of the data, it is difficult to evaluate and appreciate the discussion and conclusion. Besides, the paragraph at the end of the manuscript lines 376-382, raise some perplexities. 

Author Response

The authors of the article are grateful to the reviewers for valuable comments that made it possible to eliminate inaccuracies and make the article better.

Reviewer 2

We have made changes to the "Results" section, divided into additional sub-items:

3.1. Metabolic features of saliva in lung cancer depending on the presence/absence of COPD

3.2. Influence of histological type of lung cancer on metabolic indicators of saliva in patients with COPD

3.3. Biochemical markers of saliva in patients with lung cancer, non-cancerous pathologies of the lung and control group depending on the presence/absence of COPD

3.4. Predictive value of saliva biochemical parameters taking into account the presence/absence of COPD

Some explanations on the choice of a non-parametric method should be given. 

We have added additional information to the Materials and Methods section: «At the preliminary stage, the character of distribution and homogeneity of dispersions in groups was checked. According to the Shapiro-Wilk test, the content of all determined parameters does not correspond to the normal distribution (p<0.05). The test for the homogeneity of variances in groups (Bartlett's test) allowed us to reject the hypothesis that variances are homogeneous across groups (p=0.00017). Therefore, nonparametric statistical methods were used to process the experimental data.»

The description of the PCA factor planes is approximate. It is difficult to read the various factor planes. No indications of the criteria used to determine intensity, orientation, as well as color of the arrows, is reported. Reading the text, it seems there are incongruences. 

We have added the following information in the captions to Figures 1 and 3. «The color of the arrows on the correlation circle changes from blue (weak correlation) to red (strong correlation) as shown on the color bar. The orientation of the arrows characterizes positive and negative correlations (for the first principal component, we analyze the location of the arrows relative to the vertical axis, for the second principal component - relative to the horizontal axis).»

A comment on how the H factor has been used to analyze the data would be useful. 

We have added additional information to the Materials and Methods section: «The Kruskal-Wallis test is a nonparametric alternative to one-dimensional (intergroup) ANOVA. It is used to compare three or more samples, and tests null hypotheses, according to which different samples were taken from the same distribution, or from distributions with the same medians. With a high significance of the Kruskal-Wallis test (H), the characteristics of different experimental groups significantly differ from each other (p<0.05).»

In the legend of Fig 1 it is written "factor plane (b, e, f)" that is incorrect, and similar errors are in other figures 

Corresponding corrections were made in captions to Figures 1 and 3.

Describing the correlation circle of Fig 3 the authors state that "as well as patients with lung cancer, regardless of the presence/absence of COPD" are above the horizontal axis, incorrect. 

We have corrected the phrase in the article: «As seen in Fig. 3, the horizontal axis divides the study groups into a control group and a comparison group without COPD (above the axis), and patients with lung cancer, regardless of the presence / absence of COPD (below the axis) and a comparison group with COPD (below the axis).»

Fig 4 reports understandable indications only for curve in inset a and b, for the other nothing is reported. Besides the number "n" reported at the end of any curve, and that I assume it corresponds to the number of cases, varies from tenth to hundreds, making any comparison dubious. A comment on size and position of the rectangles is necessary. 

n is the number of patients in each group. We have corrected the captions for all tabs in the figure, divided it into 2 parts (Figure 4 - tabs a and b; Figure 5 - tabs c, e, f). In some cases, the number of patients in groups is really small, but this does not affect the correctness of the calculation, but reduces the statistical reliability. The results obtained should be regarded as preliminary; we have made this correction to the text.

Fig 2 nothing is said to facilitate the reading of the various graph 

We added the following information before Figure 2: «For a visual representation of how the indicators change while taking into account the histological type of lung cancer and the severity of COPD, diagrams of the intensity of changes are plotted in comparison with the corresponding groups without COPD (Fig. 2). To plot the diagram, only those biochemical indicators of saliva were selected, the change in which is statistically significant according to the Kruskal-Wallis test (Table 3). As can be seen from Fig. 2, for all indicators except LDH and α-amylase, a wide scatter of data is observed. Changes in these indicators against the background of COPD of varying severity most reliably show the differences between histological types of lung cancer.»

In the caption to Figure 2, added legend information. «The rectangles show the mean (line) and median (cross), individual points show values outside the interquartile range. The same letters (a, b, c, d, e, f) denote groups, the differences between which were not revealed (α = 0.05). The dotted line represents the zero level (LnRR=0). Significance of difference from 0 was checked using the t-test of one sample and indicated by numbers ("0": not significant; "1": p<0.05; "2": p<0.01; "3": p<0.001).»

Overall, due to the confusing description of the data, it is difficult to evaluate and appreciate the discussion and conclusion. Besides, the paragraph at the end of the manuscript lines 376-382, raise some perplexities. 

We have made the appropriate changes to the text: «The limitations of the study are associated with the absence of patients with severe COPD (GOLD III and GOLD IV) and with the absence of patients with COPD without other pulmonary pathologies (control group). These cases are extremely rare in our clinical practice.»

Round 2

Reviewer 2 Report

I expected a profound revision of the work, instead the authors partially responded to the points that I raised indicating that they were only some of the many problems of the work.

The title of the manuscript refers to lung cancer, however because the authors analyze adenocarcinoma and squamous cell histological types I assume the cases considered are non-small cell lung cancer. As for the neuroendocrine type, I assume that the work refers to large cell neuroendocrine carcinoma, that is reported as a rare pulmonary tumor and with features of both small cell lung carcinoma and non-small cell lung carcinoma. The mix of these three types of lung cancer might be one of the factors responsible for a certain coarseness of the data as also highlighted by the authors. Some preliminary comments would have been expected.

With respect to data analysis, the paper is a cumbersome mixture of advanced statistical techniques (PCA, Kaplan Meier regression) with poor support to the reader and hasty writing (e.g. the use of Bartlett test to assess homoschedasticity of non-normal data, the description of the applied nonparametric statistical tests, the mention of  the Kruskal Wallis test as if intended to compare sample medians ….. etc.).

Pairwise comparisons appear to be carried out in an ambiguous manner (e.g.  Table 2, or the quoted but not referred Mann-Whitney test) and this renders weak and not reader-friendly the connection between statistical data analysis and clinical conclusions.

The impact assessment of the so-called multiple comparison problem, with 34 variables, a number of groups varying from n=3 to n=5, and a significance level of 0.05 and 0.10 (e.g.  in Table 2) should have been commented, at least, against the risk of the type I error. Alternatively, a more appropriate technique ought to be selected (e.g., Bonferroni adjustment).

No considerations were made about statistical power, whereas nonparametric regression was used for prediction with a sample size as low as n=7 in figure 5c.

Overall, this referee finds that there are limitations in the analysis of the data presented, that preclude a meaningful conclusion from being reached.

The English form remains poor, there are still typing errors, some figure legends are either not complete or inappropriately written.

Author Response

The authors of the article are grateful to the reviewers for valuable comments that made it possible to eliminate inaccuracies and make the article better.

I expected a profound revision of the work, instead the authors partially responded to the points that I raised indicating that they were only some of the many problems of the work.

We made significant changes to the text of the article, structured the results section, completely rewrote some points. We believe that further changes distort the original author's structure of the work.

The title of the manuscript refers to lung cancer, however because the authors analyze adenocarcinoma and squamous cell histological types I assume the cases considered are non-small cell lung cancer. As for the neuroendocrine type, I assume that the work refers to large cell neuroendocrine carcinoma, that is reported as a rare pulmonary tumor and with features of both small cell lung carcinoma and non-small cell lung carcinoma. The mix of these three types of lung cancer might be one of the factors responsible for a certain coarseness of the data as also highlighted by the authors. Some preliminary comments would have been expected.

We have added a detailed description of the NEC patient population: «The NEC group included 16 patients with a diagnosis of typical and atypical carcinoid (low grade G1 + G2) and 45 patients with small cell lung cancer and 7 patients with large cell lung cancer (high grade G3).»

We devoted a separate section to comparing different histological types of lung cancer with and without COPD and showed that there are few significant differences between groups. Undoubtedly, it would be more correct to consider each subtype separately, however, since our study is the first such study on saliva, in the future we will consider this issue in more detail. The results of this work can be considered preliminary; we have made a corresponding change in the title of the article.

With respect to data analysis, the paper is a cumbersome mixture of advanced statistical techniques (PCA, Kaplan Meier regression) with poor support to the reader and hasty writing (e.g. the use of Bartlett test to assess homoschedasticity of non-normal data, the description of the applied nonparametric statistical tests, the mention of  the Kruskal Wallis test as if intended to compare sample medians ….. etc.). Pairwise comparisons appear to be carried out in an ambiguous manner (e.g.  Table 2, or the quoted but not referred Mann-Whitney test) and this renders weak and not reader-friendly the connection between statistical data analysis and clinical conclusions.The impact assessment of the so-called multiple comparison problem, with 34 variables, a number of groups varying from n=3 to n=5, and a significance level of 0.05 and 0.10 (e.g.  in Table 2) should have been commented, at least, against the risk of the type I error. Alternatively, a more appropriate technique ought to be selected (e.g., Bonferroni adjustment).

We have corrected the description of statistical methods: «The Kruskal-Wallis test is a nonparametric alternative to one-dimensional (intergroup) ANOVA. It is used to compare three or more samples. With a high significance of the Kruskal-Wallis test (H), the characteristics of different experimental groups significantly differ from each other (p<0.05). Using the Kruskal-Wallis test, we compared several groups (3 groups in Tables 2 and 3; 5 groups in Table 4) and selected indicators whose change was significant at p<0.05. These indicators were subsequently used for the principal component analysis (PCA). In addition, we included indicators in the PCA analysis, the values of which differ at the 0.10 significance level. In the case of 0.05<p<0.10, the limit of significance is slightly exceeded, which means that there is a tendency towards the manifestation of a pattern. If a significant pattern is identified, to identify groups that are significantly different from each other, it is necessary to test all groups in pairs. The Mann-Whitney test was used only for pairwise comparison of the differences between the COPD and non-COPD groups in different histological types of lung cancer (Figure 2); in all other cases, the Kruskal-Wallis test was used to compare the groups.»

The Bonferroni correction should be used when several pairwise comparisons are used at the same time, however, in our case, we compare only the values with COPDI and COPDII for different histological types with the corresponding value NO COPD (Figure 2). At the same time, we clarify that the differences correspond to 3 levels of significance: 0.05, 0.01 and 0.001, which excludes the possibility of a Type I error.

No considerations were made about statistical power, whereas nonparametric regression was used for prediction with a sample size as low as n=7 in figure 5c.

We have provided the following information: «Some groups contain a small number of patients; therefore, the differences are not statistically significant and can be considered as preliminary data requiring verification on more representative samples of patients».

The results of this work can be considered preliminary; we have made a corresponding change in the title of the article.

Overall, this referee finds that there are limitations in the analysis of the data presented, that preclude a meaningful conclusion from being reached.

The analysis of the data was carried out in accordance with the requirements of statistics: for paired comparison - the Man-Whitney test, for the comparison of several groups - the Crasskel Wallis test, to describe the relationship between the parameters - the PCA method, the analysis of survival - the Kaplan Meier method. The normal distribution was checked. The team of authors includes a data analyst (Solomatin Denis).

The English form remains poor, there are still typing errors, some figure legends are either not complete or inappropriately written.

Corrections have been made to the text.